# Peptidoglycan recognition protein PGRP-5 is involved in immune defence and neuro-behavioral disorders in zebrafish embryos

**Xi Li[1]◉, Guanghua Xiong[2]◉, Manni Luo[2]◉, Siwan Mao[2], Ruiying Zhang[2], Ziwei Meng[2], Juan Li[1], Xinjun Liao◉[1] ***

1 Department of Stomatology, Affiliated Hospital of Jinggangshan University, Clinical Medical Research Center of Jinggangshan University, College of Life Sciences, Jinggangshan University, Ji'an, Jiangxi, China, 2 College of Biology and Food Engineering, Key Laboratory of Embryo Development and Reproductive Regulation of Anhui Province, Fuyang Normal University, Fuyang, Anhui, China

◉ These authors contributed equally to this work.
* xinjun_liao2022@126.com

**Data Availability Statement:** All relevant data are within the manuscript and its Supporting Information files.

**Funding:** This research was funded by the National Natural Scientific Foundation of China (82160048,

## Abstract

Peptidoglycan recognition proteins (PGRPs) are the evolutionarily highly conserved class of pattern recognition receptors, however, their functions on the innate immune system and neuro-inflammatory response in aquatic organism are still poorly understood. In this study, we systematically investigated the molecular functions of PGRPs in zebrafish embryos. Firstly, we identified three PGRPs in zebrafish and phylogenetic analysis suggested that DrPGRP-5 was a novel member of the PGRP superfamily in evolution. Secondly, the endogenous mRNA levels of DrPGRP-5 were highly expressed in brain and muscle while significantly down-regulated in liver and egg at 72 hpf in zebrafish embryos. Thirdly, the mRNA levels of DrPGRP-5 were greatly elevated after 6 h of *E. coli* infection but reached its highest value at 24 h after *M. luteus* stimulation. Moreover, knock-down DrPGRP-5 could significantly reduce the pro-inflammatory cytokines such as *TNF-α*, *IL-1β* and *IL-6*, but increased the expression of anti-inflammatory cytokine *TGF-β*. On the other hand, the loco-motor behavior abilities and the antioxidant enzyme activities such as CAT and SOD were obviously decreased under the DrPGRP-5 KD conditions. Finally, incubation of zebrafish embryos with anti-inflammatory and neuroprotective agents (10 μM Minocycline) can partially rescue the DrPGRP5-regulated locomotor behavior. Taken together, our data suggested that zebrafish PGRP-5 is involved in the innate immune defenses and regulated the neurobehavior and neuro-inflammation, which may provide new strategies for the treatment of neuro-inflammatory diseases in the aquatic organisms.

## 1. Introduction

The innate immune system recognizes microorganisms through highly conserved pattern recognition receptors (PRRs) in evolution from insects to mammals [1,2]. These PRRs can identify specific components of external pathogenic microorganisms, namely pathogen associated

82460319), Anhui Natural Science Foundation Project (2308085MH265), Anhui Science and Technology Research Project of the Education Department (2024AH040205), Anhui Excellent Talents Support Program for Universities (YQYB20230170), Jiangxi Science and Technology Research Project of the Education Department (GJJ2201611), PhD Initiation Project of Fuyang Normal University (KYQD20230004) and PhD Initiation Project of Jinggangshan University (JZB2016).

**Competing interests:** The authors have declared that no competing interests exist.

molecular patterns (PAMPs), thereby activating relevant signaling pathways to initiate natural immune responses and effectively eliminate pathogens [3,4]. To date, numerous PRRs have been reported such as peptidoglycan recognition proteins (PGRPs), Gram-negative binding proteins (GNBPs), and Toll-like receptors (TLRs) [5,6]. However, the immune and other biological functions of PRRs in aquatic organisms such as zebrafish has not been fully studied yet.

PGRPs are a class of evolutionarily conserved molecules involved in the immune response against pathogens, which can activate the Toll pathway or immune deficiency (IMD) pathway to produce antibacterial or antiviral effects [7,8]. PGRPs can recognize Lys- and DAP-type peptidoglycans in bacterial walls in various species from insects to mammals, which possessed the ability to trigger the prophenoloxidase (PPO) cascade [9,10]. The C-terminal PGRP domain, which is homologous to bacterial type 2 amidases and bacteriophages, is a conserved feature of members of the PGRP family [11,12]. At present, studies on PGRPs have been explored in both invertebrates and vertebrates, and there has been several research on their expression and function in insects, fish and mammals [13,14]. However, PGRPs were not found in lower organisms such as nematodes and plants [15]. The discovery of PGRPs suggests that PGRPs may become a powerful link between invertebrate and vertebrate immunity [16,17]. The unique conserved domain of PGRPs determines its important role in both vertebrates and invertebrates, and the diversity of PGRPs in species and its structure also indicates their multiple functions in innate immunity and other biological regulatory effects.

Nowadays, with the development of fisheries and aquaculture, the fish are facing the survival challenge due to pathogenic microorganisms in the aquatic environment [18]. The abuse of a large number of antibiotics has led to an increasing resistance of bacteria to antibiotics, posing a great threat to the development of fisheries economy and human health [19,20]. Studying PGRPs in fish is crucial as they are the most important PRRs molecules for resisting bacterial and viral infections in aquatic ecosystems [21,22]. Almost all PGRPs contain one or two PGRP conserved domains composed of 165 amino acid residues at the carboxyl or C-terminus, which has a specific binding site for the cell wall acyl peptide fragment of bacterial PGN [23,24]. In teleosts, several PGRPs have been identified in zebrafish and rockfish [25,26]. Zebrafish PGRPs, which mediate numerous intracellular signaling pathways including TLR signaling and cell death, have the amidase activity and broad-spectrum bactericidal activity in contrast to insect and other mammalian homologs [27,28]. However, the spatiotemporal expression patterns of each type of PGRP in zebrafish and its dynamic changes under pathogen stimulation are currently unclear.

Recent studies had found the role of peptidoglycans and their recognition molecules in neural development and behavior in neurons and glial cells [29]. PRRs recognize conserved microbial molecular features such as peptidoglycans on bacterial surfaces and have become potential key regulatory factors for gut microbiota-brain interactions in the placenta and developing brain [30]. Evidence has been found that PGN and its sensing molecules play multiple important roles beyond innate immunity, which extend to neural development and behavior in autism spectrum disorder (ASD) [31]. However, little is known about whether PGRP is involved in neuro-inflammation or behavioral abnormalities in zebrafish embryos.

In order to gain insight into the function of zebrafish PGRPs in innate immunity and neural behavior, we firstly identified three zebrafish PGRP genes, including a short PGRP with amidase activity (designated as DrPGRP5) and two long PGRPs (designated as DrPGRP2 and DrPGRP6, respectively) in the present study. Then, we conducted a phylogenetic tree analysis on the zebrafish PGRP genes and homologous genes of other species. Meanwhile, the mRNA expression levels of zebrafish three PGRPs in various tissues and DrPGRP5 in response to Gram-negative and positive bacteria stimulation were also investigated by real-time qRT-PCR experiments. Besides, we also examined the dynamic expression of inflammatory genes at

different time points and their expression levels under PGRP5 gene knockdown conditions. Furthermore, we explored the behavioral changes and neuro-related enzyme activities of DrPGRP5 knockdown on zebrafish, and validated the regulation of DrPGRP-5 on neuro-behavior through pharmacological rescue experiments. Altogether, the information will help to elucidate the new biological functions of PGRPs on neuro-inflammatory effects in fish.

## 2. Materials and methods

### 2.1. Zebrafish maintenance

The wild type AB zebrafish were purchased from China Zebrafish Resource Center (CZRC, http://www.zfish.cn/) to ensure a clear genetic background and no pathogen infection. In our facilities, the adult zebrafish were grown and kept in recirculating water tank at a temperature of 28±1°C with a photoperiod of 14 h of light and 10 h of darkness. Brine shrimp were given twice daily to the zebrafish. Within spawning boxes, male and female adult fish were segregated overnight into different areas (1:1 or 2:1 ratio). Spawning began the following morning after the baffle was taken out. Embryos were then gathered in egg water and prepared for usage. After treatment, all zebrafish were immediately anesthetized by 0.02% MS-222 and sacrificed for various tissues collection. In addition, all experiments were approved by the Ethics committee and carried out according to approved guidelines for animal welfare (IACUC protocol# JGSU-IACC-202203002). We strictly adhere to biosafety rules of Jinggangshan University and all methods are reported in accordance with ARRIVE guidelines.

### 2.2. Multiple sequences alignment and phylogenetic analysis

The BLAST method was used to examine the potential candidates for the zebrafish PGRP genes against the NR database with a cut-off E-value of 0.01. Pfam (http://pfam.xfam.org/) and SMART (http://smart.embl-heidelberg.de/) were used to further confirm the deduced amino acid sequences in order to find conserved domains needed for particular functions. Sequences from other representative species, such as Homo sapiens, Mus musculus, and Oryzias latipes, were then matched with those from zebrafish PGRPs. With the use of the BLOSUM series of weight matrices and the ClustalX program, multiple sequence alignments were carried out [32]. The neighbor-joining (NJ) approach was used using MEGA 6 software to create a phylogenetic tree, and 1000 bootstrap repetitions were used to assess the branching's dependability [33].

### 2.3. The spatial and temporal expression patterns of zebrafish PGRPs

Total RNAs from various tissues in three-month adult zebrafish were extracted using TRIzol reagent (Thermo Fisher Scientific, CAS no. 15596018, USA). To create the first-strand cDNA, 1 μg of total RNA was used and reverse transcribed with M-MLV reverse transcriptase from Promega (USA). The reverse transcription reaction was heated for 5 minutes at 95°C after being incubated at 42°C for 1 hour. The SYBR Green qPCR Mix (Biosharp, CAS no. BL698A, China) was used for the qRT-PCRs. The PGRP2, PGRP5 and PGRP6 genes from different tissues were examined for the spatial and temporal expression profiles, with the expression of the β-actin gene serving as an internal control.

### 2.4. Bacterial challenge and embryo exposure experiments

We selected the Gram$^+$ and Gram$^-$ strains of Escherichia coli (*E. coli* ATCC 27325) and Micrococcus luteus (*M. luteus* ATCC 10240) and perform activation culture on LB medium until the logarithmic growth phase. Adjust the bacterial concentration to the desired level (e.g. 1 x 10$^7$

CFU/mL) using spectrophotometry for subsequent experiments. After 6 hours of zebrafish embryo development, the above two bacterial solutions were added into the culture medium and incubated for different time periods. Then, the total RNA of zebrafish embryos at 72 hpf was extracted in advance and the mRNA expression level of PGRP5 gene in various tissues were detected. Simultaneously set up a control group without bacterial inoculation to evaluate the specific impact of bacterial challenge on embryos. Besides, dead embryos and larvae were removed daily, and fresh solution was changed every day. Finally, the survival rate and hatching rate of zebrafish embryos in each group was statistically analyzed.

## 2.5. Morpholino knockdown and qRT-PCRs

Using morpholinos, we aimed to silence the DrPGRP5 gene in zebrafish embryos. Briefly, GeneTools (OR, USA) created the antisense morpholino oligonucleotide (5'-ATGTTATCTTC TTACTTGTAACCGA-3') to selectively target the zebrafish PGRP5 mRNA (GenBank No. NM_001044321). As a negative control, the standard vivo-morpholino from Gene Tools (5'-CCTCTTACCTCAGTTACAATTTATA-3') was employed. For microinjection, morpholinos were diluted to the necessary concentrations in 1 nl of 1% phenol red-containing Danieau's solution, and then one nanoliter (nl) was injected into the yolk region of the embryo at the one- to two-cell stage.

The ABI StepOne Plus system (Applied Biosystems, USA) was used to measure the relative expression levels of the genes involved in inflammation. The heat cycle was as follows: denaturation for 2 min at 95°C, then 40 cycles each lasting 15 s at 95°C, 15 s at 60°C, and 30 s at 72°C for extension. The examination of the dissociation curve was used to determine the purity of the generated PCR products. The relative expression values of a particular gene were analyzed using the comparative Ct method and normalized to an endogenous control β-actin (triplicates for each treatment). Table 1 contains a list of the primer sequences utilized in this experiment.

## 2.6. Behavioral analysis in zebrafish embryos under PGRP5 knock-down conditions

After 72 hpf of medication treatment, a zebrafish behavioral experiment was conducted. In a nutshell, 12 zebrafish larvae from each AlCl3 or CYP-treated group were put into a 24-well plate, one larva per well, with 500 μL of embryo media, and kept inside the dark box of the zebrafish behavioral instrument for around 15 min to adjust before engaging in behavioral tracking. Using the DanioVision Observation Chamber System (Noldus IT, Netherlands), the free-swimming activities of zebrafish were captured in response to a 20-min light-to-dark photoperiod stimulation. Using the EthoVision XT software, the movies were evaluated for behavioral

**Table 1. Sequences of primer pairs used in the real-time quantitative PCR reactions.**

| Gene name | Forward primer | Reverse primer | Accession No. |
|---|---|---|---|
| PGRP2 | ATTGCCCGAGCATCATTCCT | GCTACACCATACCCCACGTT | NM_001045166 |
| PGRP5 | CGCTGATATGGACGGACACA | AGCACAAAATTGGGTCGCAC | NM_001044321 |
| PGRP6 | GTCTTCGTAAAGCCTCCGGT | GATGGTCAGACGAGGCCATT | NM_001045222 |
| IL-1β | CGTGAAGTGAACGTGGTGGA | GTACGAGATGTGGAGACGTGG | AY340959 |
| IL6 | CCTCAAACCTTCAGACCGCT | GAACAGGATCGAGTGGACCG | NM_001261449 |
| TNF-α | CAATCCGCTCAATCTGCACG | TACAGATGTGTTGGCGGCAC | NM_212859 |
| TGF-β | GTCCGAGATGAAGCGCAGTA | TCAAATGAGAGCCAGCGGTT | NM_001044759 |
| β-actin | CGAGCAGGAGATGGGAACC | CAACGGAAACGCTCATTGC | AF057040.1 |

changes in the behavioral metrics taken from behavioral trajectory records, such as the overall distance and average speed in each concentration group. Minocycline hydrochloride (CAS: HY-17412A) was purchased from MedChemExpress (MCE) Co., Ltd (New Jersey, USA) and used for pharmacological experiments.

## 2.7. Detection of neural and antioxidant enzyme activities in zebrafish

With a few modest adjustments to the manufacturer's instructions, the levels of acetylcholinesterase (AchE), catalase (CAT), superoxide dismutase (SOD), and reactive oxygen species (ROS) concentrations were tested using emzyme activity kits. In a nutshell, 40 zebrafish larvae from each group were collected and homogenized in 500 μL of ice-cold PBS on ice. To obtain the supernatant, the homogenate was centrifuged at 2500 x g for 10 min at 4°C. The BCA protein assay kit (Cat No. P0010, Beyotime Biotechnology Co., Ltd., China) was used to measure the protein concentration. Following that, the microplate reader (SpectraMax iD3, MD, USA) is used to measure the absorbance, and the associated enzyme activities were calculated using the provided formula.

## 2.8. Statistical analysis

SPSS 19.0 was used to analyze all of the experiment outcomes in the current study. The mean ± SEM (Each value contains at least four biological replicates, and each biological replicate contains three technical replicates) were calculated and used to depict the statistics. The unpaired student t-test or one-way ANOVA followed by the Tukey multiple comparison test were used to establish the statistical significance. $p < 0.05$ was seen as significant for all tests and denoted with the *, whereas $p < 0.01$ was regarded as extremely significant and denoted with the **.

# 3. Results

## 3.1. Sequence analysis of zebrafish PGRPs

In order to identify PGRP genes in the zebrafish genome, we searched the NCBI GenBank database for genes homologous to human PGRPs. As a result, we totally identified three PGRP genes (namely PGRP2, PGRP5 and PGRP6) in the zebrafish GRCz11 genome by using BLAST comparsion. It is worth mentioning that zebrafish PGRP5 had the highest homology to mammalian PGRP-1 when compared to other zebrafish PGRPs. Meanwhile, multiple sequence alignments indicated that zebrafish PGRPs were highly homologous to other PGRPs from various species such as ricefish (*Oryzias latipes*), human (*Homo sapiens*) and mouse (*Mus musculus*) (Fig 1A and S1 File). All zebrafish PGRPs share one N-acetylmuramoyl-L-alanine amidase domain at their C-terminus with the majority of vertebrate and invertebrate PGRPs, which have 61% (DrPGRP2), 52% (DrPGRP5) and 70% (DrPGRP6) conserved identities when compared with ricefish, human and mouse, respectively (Fig 1B). Furthermore, all zebrafish PGRPs are predicted to have amidase activity because they have well conserved amidase catalytic sites (His 316, Tyr352, His427 and Cys436 of PGRP2; His 98, Tyr132, His206 and Cys214 of PGRP5; His 359, Tyr395, His470 and Cys478 of PGRP6), which is known to be critical for $Zn^{2+}$ and amidase activity in zebrafish.

A phylogenetic tree was created based on the entire lengths of the PGRPs in zebrafish using the neighbor-joining method and 1000 bootstrap tests. Our results showed that zebrafish DrPGRP2 and DrPGRP6 are most similar to ricefish OlPGRP2 and OlPGRP6, which suggested that the two species have the highest homology in evolution (Fig 2). Interestingly, DrPGRP5 and OlPGRP5 was separately clustered together with human and mouse PGRPs, which demonstrated that zebrafish PGRP5 displayed an unique evolutionary characteristics.

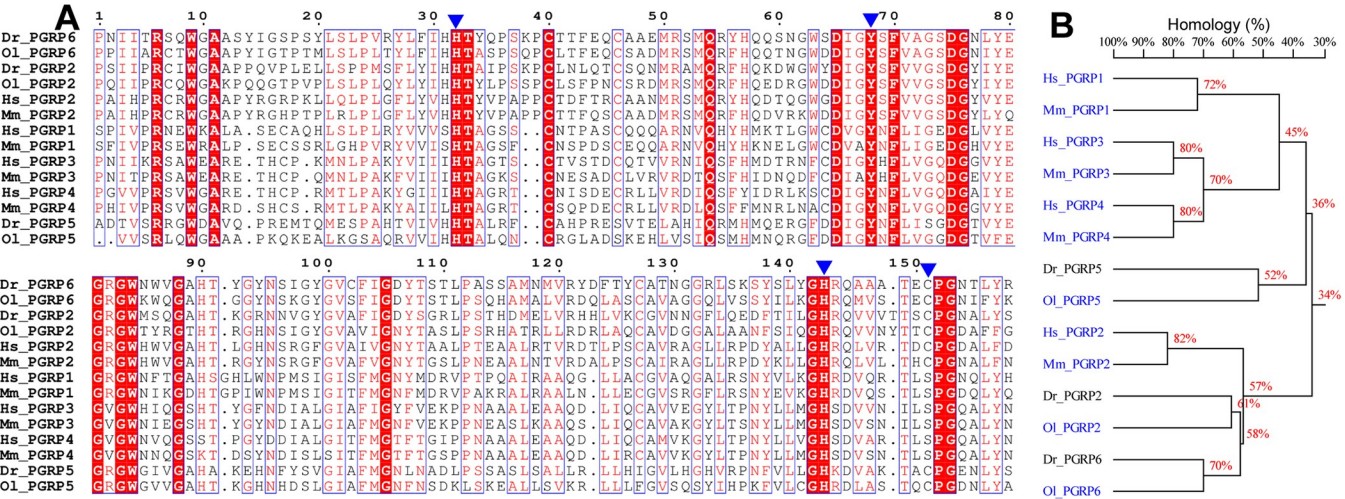

**Fig 1. Multiple alignments of C-terminal amino acid sequences contained conserved amidase domains in zebrafish (*Danio rerio*, *Dr*), ricefish (*Oryzias latipes*, *Ol*), human (*Homo sapiens*, *Hs*) and mouse (*Mus musculus*, *Ms*) PGRPs.** (A) The multiple sequence alignment of PGRPs was presented. The conserved amino acid residues were shaded in red, and similar amino acids were marked with red characters. (B) The homology percentage of PGRPs gene sequences was presented. The protein sequences were listed as followed: Zebrafish (PGRP2: NP_001038631; PGRP5: NP_001037786; PGRP6:NP_001038687), ricefish (PGRP2: NP_004065551; PGRP5: NP_004075905; PGRP6:NP_004071889), human (PGRP1: NP_005082; PGRP2: NP_443122; PGRP3: NP_443123; PGRP4: NP_065126) and mouse (PGRP1: NP_033428; PGRP2: NP_067294; PGRP3: NP_997130; PGRP4: NP_997146).

## 3.2. Tissue differential expression patterns of DrPGRPs

In order to further explore the structural characteristics of PGRPs in zebrafish, we identified the putative domains in each PGRP. The results revealed that each PGRP contains a conserved

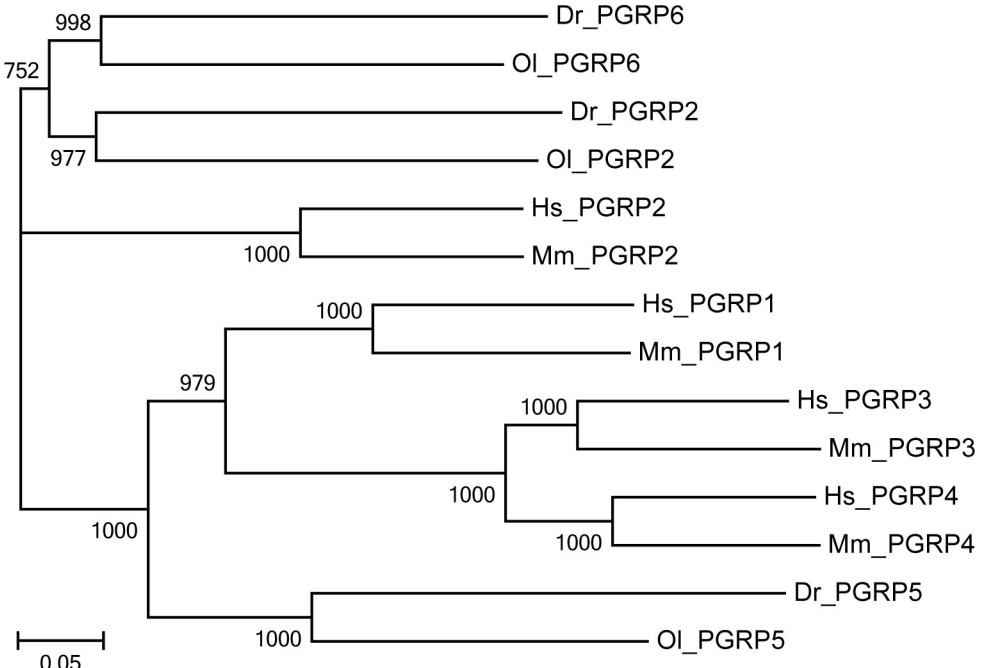

**Fig 2. Phylogenetic tree analysis of DrPGRPs with other known PGRPs form various species.** Phylogenetic tree was obtained from a CLUSTALW alignment and MEGA6 Neighbor-joining of 14 sequences. The number at each node indicates the bootstrap analysis from 1000 replicates.

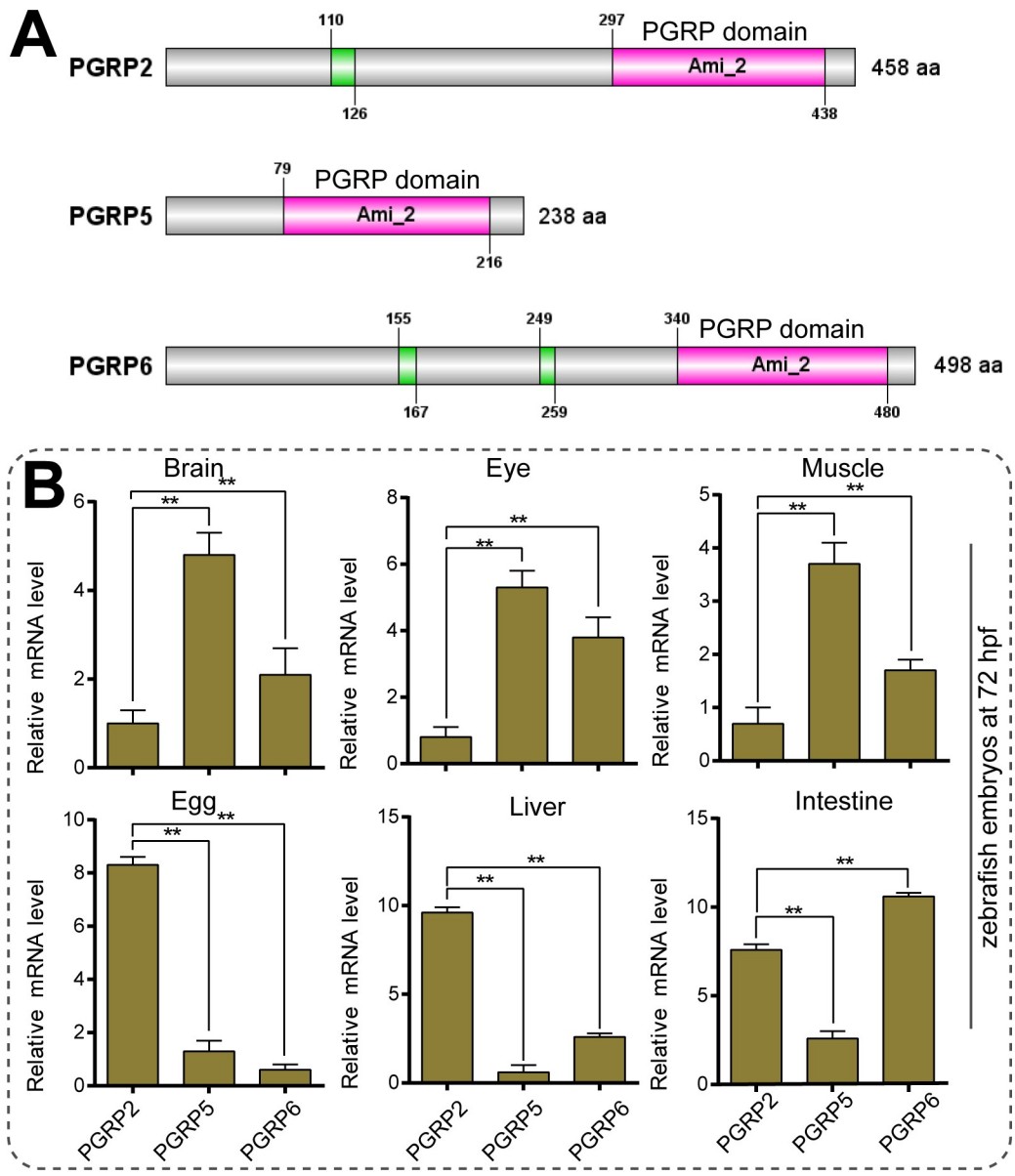

**Fig 3. Structural characteristics and expression patterns of zebrafish PGRPs genes.** (A) Schematic representations of the zebrafish PGRP structures. Characteristic domains and lengths of the amino acid sequences are indicated. The green boxes represent the low complexity region, and pink boxes stand for PGRP domains that contained a conserved N-acetylmuramoyl-L-alanine amidase (ami_2) region. (B) The mRNA expression levels of zebrafish PGRPs in different tissues. The relative transcript levels of DrPGRP2, 5, 6 in brain, eye, muscle, egg, liver and intestine were presented. The values in each group were presented as means ± SEM (Each value contains at least four biological replicates, and each biological replicate contains three technical replicates). For all experiments, $^*p < 0.05$; $^{**}p < 0.01$.

N-acetylmuramoyl-L-alanine amidase (ami_2) domain in the C-terminal of full-length sequences (Fig 3A). Structural analysis suggested the ami_2 domain is conserved through evolution, and these proteins are predicted to be amidases potentially serving as peptidoglycan receptors triggering immune signaling pathways.

To investigate the tissue-dependent expression patterns, we performed real time-quantitative PCRs using gene-specific primers for DrPGRP2, DrPGRP5 and DrPGRP6 (Fig 3B). DrPGRP5 had the high expression in the brain, eye and muscle but a low expression in the egg

and liver. On the other hand, DrPGRP2 and DrPGRP2 were significantly high expressed in the intestine tissues. These results indicated that the expression profiles of the three DrPGRPs in zebrafish are differentially regulated despite the high conservation of these proteins.

### 3.3. Bactericidal activity of zebrafish PGRP-5 gene

Since some PGRPs from other vertebrates have been shown the potential antibacterial activity, we wondered whether zebrafish PGRPs have bactericidal activity. Our results suggested that the expression of PGRP-5 showed different expression characteristics against Gram-negative bacteria (*E. coli*) and Gram-positive bacteria (*M. luteus*). Firstly, the survival rate significantly decreased over time under two different bacterial stimuli, especially at 72 h, with *E. coli* showing a more severe decline than *M. luteus* (Fig 4A). In addition, the hatching rate at 24 hpf has also decreased under the stimulation of two kinds of bacteria, but *M. luteus* has decreased more than *E. coli* (Fig 4B).

On the other hand, the mRNA expression levels of DrPGRP-5 were significantly increased at 6 hours after infection of *E. coli*, and then returned to normal level at 24 hours. After that, the expression level decreased with the infection time (Fig 4C). On the contrary, the expression of DrPGRP-5 was up-regulated after infection of *M. luteu*s, which reached the peak at 24 hours post infection (Fig 4D). From above results, it is demonstrated that zebrafish PGRP-5 plays different functions in against Gram-positive and negative bacteria.

### 3.4. The inflammatory cytokines were differentially regulated upon PGRP-5 KD conditions

As an important pattern recognition receptor, PGRP can regulate the expressions of many downstream effector genes. Therefore, we wonder to explore whether the classic inflammatory cytokines were regulated by PGRP-5. Firstly, we blocked the zebrafish PGRP-5 gene by vivo-morpholinos. Based on the pre-experimental results, the transcriptomic expression of PGRP-5 could be reduced by approximately 80% using morpholinos by quantitative real-time PCR analysis (S1 Fig). Our results suggested that majority of the inflammatory genes were significantly decreased after PGRP-5 KD conditions. For instance, IL-1β is a component of systemic inflammation, and its expression was dose-dependently reduced from 6 to 72 hours, but there was no discernible change when PGRP5-KD was present (Fig 5A). On the other hand, IL-6, a leukocytic endogenous mediator, has a much lower expression level after infection. However, under PGRP5-KD circumstances, IL-6 mRNA clearly dropped almost at the time of each infection (Fig 5B). TNF-a was a soluble cytokine that is released by a number of immune cells when they are stimulated. In the PGRP5-KD circumstances, the expression level of TNF-α gene was markedly reduced (Fig 5C). TGF-β, on the other hand, is an anti-inflammatory cytokine that controls a number of biological processes, including inflammation, cell differentiation, and embryonic development. In the PGRP5-KD circumstances, the levels of these genes significantly increased (Fig 5D). These findings collectively showed that the inflammatory genes were differently regulated in zebrafish embryos under PGRP5-KD circumstances.

### 3.5. The neurobehavioral dysfunction after PGRP5-KD in zebrafish embryos

In addition, we have further investigated the neurotoxic effects of PGRP-5 in zebrafish embryos. We used the DanioVision Observation Chamber System to evaluate the locomotor behavior of zebrafish larvae after PGRP5-KD conditions. Our results suggested that PGRP5-KD can significantly decrease the locomotion behavior of zebrafish larvae at 5 dpf,

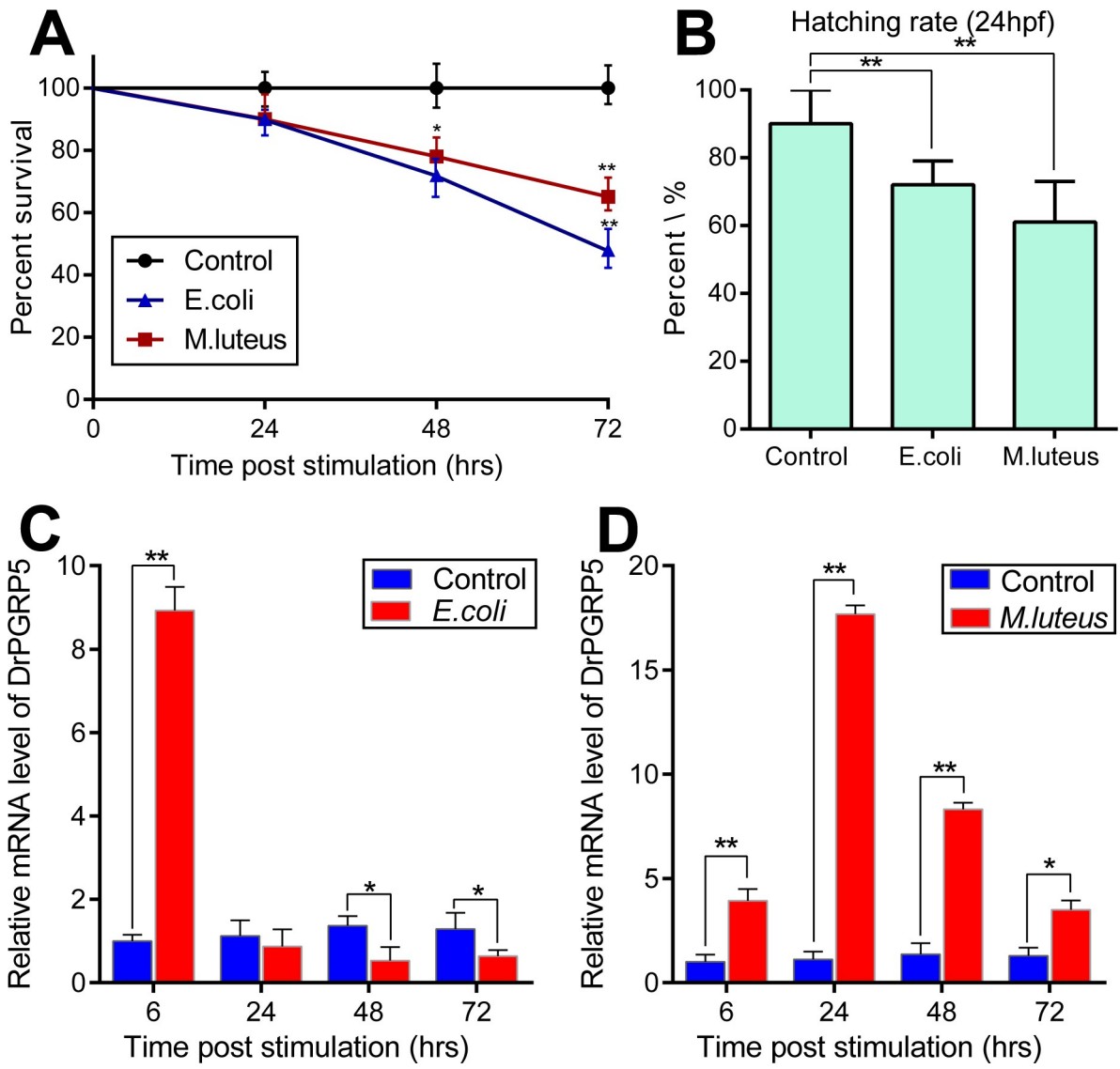

**Fig 4. Zebrafish PGRP5 exhibits differential expression characteristics under different bacterial stimuli.** (A) The survival rate of zebrafish embryos at 72 hpf after Gram-negative *E.coli* (A) and Gram-positive *M. luteus* (B) challenge. (B) The hatching rate of zebrafish embryos at 24 hpf after *E.coli* and *M. luteus* challenge. (C) The temporal mRNA expression profiles of zebrafish PGRP5 in zebrafish larvae after *E. coli* challenge. (D) The temporal mRNA expression profiles of zebrafish PGRP5 in zebrafish larvae after *M. luteus* challenge. We detected the expression of PGRP5 gene under different stimuli at time points 6, 24, 48, and 72 hours, respectively. The values in each group were presented as means ± SEM (Each value contains at least four biological replicates, and each biological replicate contains three technical replicates). For all experiments, *$p < 0.05$; **$p < 0.01$.

which were observed from both the locomotion traces and the behavioral heatmaps (Fig 6A). Besides, both the total distances moved and the mean velocity was significantly down-regulated in PGRP5-KD groups compared with the control group (Fig 6B and 6C).

To test whether the oxidative stress was regulated in zebrafish embryos after PGRP5-KD conditions, the corresponding antioxidant enzyme activities were detected in zebrafish embryos at 72 hpf. Our results suggested that the enzyme activity of CAT, SOD, and ROS was significantly decreased under PGRP5-KD group (Fig 6D). On the other hand, AcHE is a key enzyme in biological nerve conduction and plays an important role in the excitatory effect of

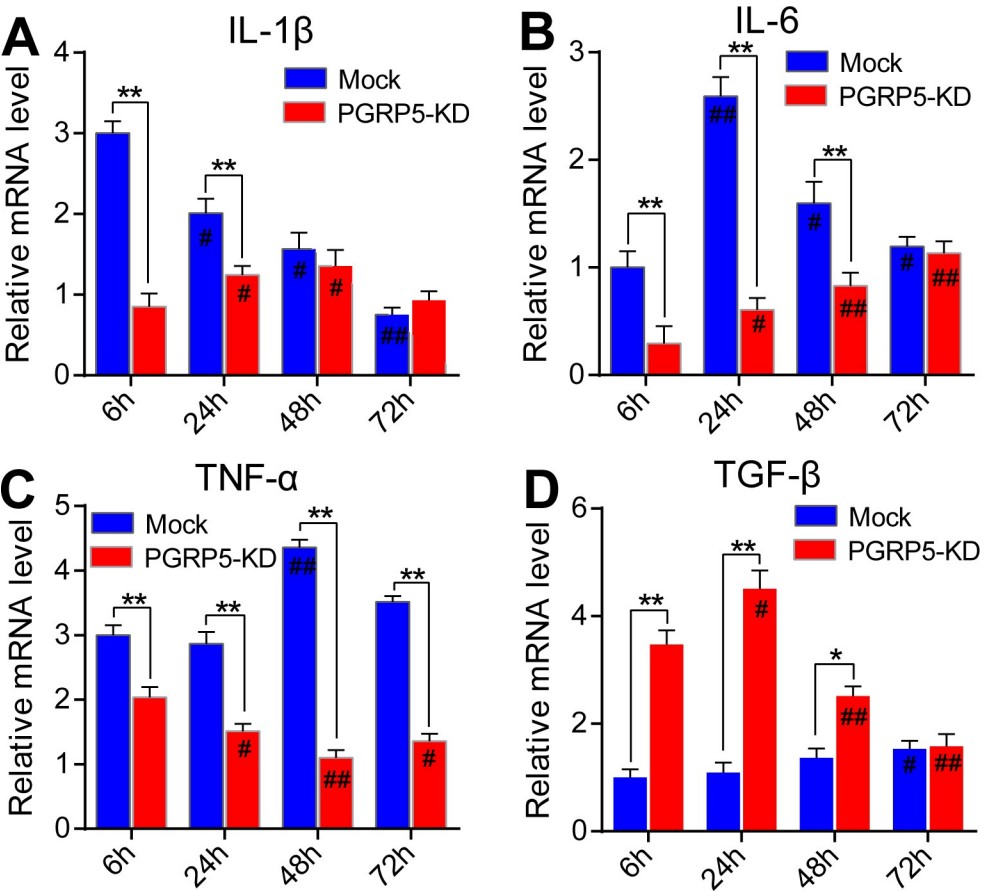

**Fig 5. The pro-inflammatory cytokines were mainly inhibited but anti-inflammatory genes were significantly activated in zebrafish embryos under PGRP5-KD conditions.** (A) The relative mRNA levels of *IL-1β* at 6, 24, 48, and 72 hpf both in the mock and PGRP5-KD conditions. (B) The relative mRNA levels of *IL-6* at 6, 24, 48, and 72 hpf both in the mock and PGRP5-KD conditions. (C) The relative mRNA levels of *TNF-α* at 6, 24, 48, and 72 hpf both in the mock and PGRP5-KD conditions. (D) The relative mRNA levels of *TGF-β* at 6, 24, 48, and 72 hpf both in the mock and PGRP5-KD conditions. The values in each group were presented as means ± SEM (Each value contains at least four biological replicates, and each biological replicate contains three technical replicates). For all experiments, $*p < 0.05$ and $**p < 0.01$ for intra-group comparison; $\#p < 0.05$ and $\#\#p < 0.01$ for inter-group comparison.

neurotransmitters on postsynaptic membrane. Our results have shown that the enzyme activities of AcHE were obviously up-regulated after PGRP5-KD conditions. These results demonstrated that the neurobehavioral effects and oxidative stress were regulated by PGRP5 in zebrafish embryos.

Furthermore, we employed the tetracycline antibiotic minocycline to treat the PGRP5-KD zebrafish and studied the characteristics of locomotor behavior in order to further assess if changes in oxidative stress will alter the neurobehavior in zebrafish. According to the findings, incubating 10 μM minocycline, which has anti-inflammatory and neuroprotective properties, can partially treat neurobehavioral disorders that display a marked improvement in locomotor behavior (Fig 6E). In addition, the minocycline-treated group's mean velocity and total moving distance of zebrafish larvae were considerably increased when compared to the control group, respectively (Fig 6F and 6G). Overall, these findings showed that by controlling neuroinflammation in zebrafish embryos, minocycline might partially reverse the PGRP5-regulated neurobehavioral impairment.

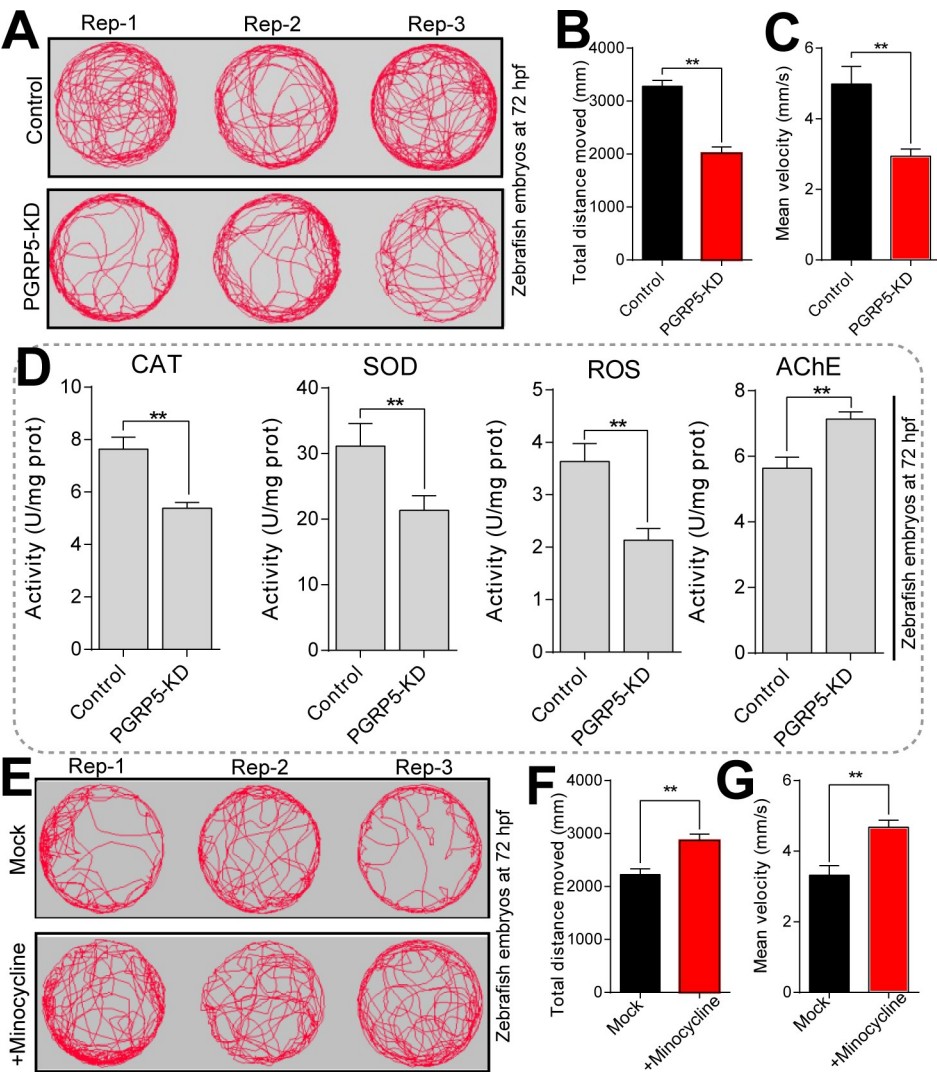

**Fig 6. PGRP5 plays an important role in neurobehavioral alteration and oxidative stress in zebrafish embryos.** (A) The movement tracks of zebrafish at 5 dpf larvae that exposed to mock and PGRP5-KD conditions. Three representative photographs are shown in each group. (B) The total distance moved of zebrafish larvae that both in control and PGRP5-KD conditions. (C) The average velocity of zebrafish larvae that both in mock and PGRP5-KD conditions. (D) Detection the contents of ROS, along with enzymatic activities of CAT, SOD and AChE in larval zebrafish that both in mock and PGRP5-KD conditions. The values are presented as means ± SEM (Each value contains at least four biological replicates, and each biological replicate contains three technical replicates). *, $p < 0.05$; **, $p < 0.01$. (E) The behavioral trajectory characteristics of zebrafish under 10 μM minocycline drug exposure. (F-G) The total distance and mean velocity of zebrafish larvae that exposed to mock and 10 μM minocycline.

## 4.Discussion

Like other vertebrates, zebrafish have both an innate and an adaptive immune system, but aquatic species rely more heavily on innate immunity than mammals do to protect themselves from bacterial infection [34,35]. Pattern recognition receptors (PRRs) and their signaling pathways have essential roles in recognizing various components of pathogens and triggering inflammatory responses that eliminate invading microorganisms and damaged cells [36]. Peptidoglycan Recognition Protein (PGRP) is one of the pattern recognition proteins that plays a significant role in the innate immunological responses of insects and humans [37]. In the

present study, we identified three PGRP homologues in the zebrafish, analyzed their expression patterns and characterized their functions in immune responses.

Phylogenetic analysis suggested that zebrafish PGRP2, PGRP5 and PGRP6 proteins are highly conserved and have one PGRP domain on the C-terminus, which is homologous to the PGRP domains of other vertebrate PGRPs including human and mouse. Furthermore, the amino acids of His98, Tyr132, His206 and Cys214 in zebrafish PGRP5, which are required for $Zn^{2+}$ binding and amidase activity. In particular, the cysteine residue of this site is indispensable for amidase activity, but is not present in human and mouse HsPGRP1, 3 and 4. On the other hand, zebrafish PGRPs are selectively expressed in a wide range of tissues including the liver, intestine, maturing oocytes and skin. Both PGRP-2 and PGRP-5 are expressed in growing embryos, and PGRP-2 is also highly expressed in the eggs. The fact that PGRPs are expressed in a variety of organs indicates that this family of proteins plays a significant role in adult zebrafish defense against bacterial infections.

It is known that amidase or bactericidal activity can be found in mammalian PGRPs [38]. It's possible that humans and zebrafish PGRPs share a similar mechanism for eliminating microorganisms. According to our findings, the brain, eye, and spleen of zebrafish exhibit high levels of PGRP-5 expression, while each kind of PGRP displayed a distinct tissue expression profile. Broad-spectrum antibacterial action of zebrafish PGRPs is demonstrated against both Gram-positive and Gram-negative bacteria. Comparable to zebrafish PGRPs, recombinant *Sebastes schlegeli* PGRPs have broad-spectrum antibacterial action according to previous studies [39]. As a matter of fact, it has been proposed that zebrafish PGRP-5 plays a role in a variety of signals transduction pathways that support immune responses as well as other biological processes like development and apoptosis [40]. Thus, it would be important to look into the precise role that zebrafish PGRP plays in innate immunity. It has been demonstrated that PRRs in vertebrates, such as TLRs and NODs, are involved in the signaling cascade that triggers the release of inflammatory cytokines [40]. According to our research, zebrafish PGRPs may mediate the innate immune defense by removing pathogenic microorganisms and lowering pro-inflammatory cytokines.

In addition, IAB and probiotic treatments promote neuronal health and influence inflammatory pathways through PGRPs-mediated neural and immune signaling in drosophila model [41]. Our results strongly suggested that zebrafish PGRP-5 had the potential to induce neurotoxicity and locomotor impairments in zebrafish larvae. The moved distance and swimming velocity of zebrafish larvae exposed to PGRP-5 knock-down conditions at 72 hpf were severely decreased when compared with those in the control group. Meanwhile, enzyme activities of antioxidant proteins including CAT, SOD, and ROS were significantly down-regulated under PGRP5-KD condtions. Besides, the up-regulation of AChE in PGRP5-KD could potentially be attributed to several factors. One possibility is that the knockdown of PGRP5 might disrupt normal cellular signaling pathways that regulate the expression of AChE and its knockdown could lead to compensatory mechanisms or imbalances in downstream signaling resulting in the increased expression of AChE. Another consideration is that PGRP5 might directly or indirectly interact with transcription factors and the absence or reduced function of PGRP5 could remove an inhibitory effect or trigger a feedback loop that promotes AChE up-regulation. In conclusion, we have identified the new role of zebrafish PGRP-5, which are highly conserved with PGRP homologues from other vertebrates. Zebrafish PGRP-5 has both amidase and bactericidal activities, and they widely regulated the neurobehavioral dysfunction and oxidative stress in zebrafish embryos. Taken together, further research is needed to understand the molecular mechanisms of zebrafish PGRP5 in other aquatic organisms.

## 5.Conclusion

In summary, our findings indicated that zebrafish PGRP5 plays an important role in immune defense and neurobehavioral disorder in zebrafish embryos. PGRP-5 is evolutionarily specific, and it is also present unique characteristics in response to bacterial infections. In addition, PGRP-5 can inhibit the locomotor behavior and modulate the neuro-inflammatory genes and oxidative stress. Taken together, our studies provided a global view of functional role of PGRP-5 in zebrafish embryos. The information will be helpful to understand the possible molecular mechanisms of PRRs in aquatic organisms.

## Supporting information

**S1 File. The full-length amino acid sequences of PGRP proteins from four homologous species.**
(DOCX)

**S1 Fig. Detection of knockdown efficiency of zebrafish PGRP5 gene by real-time quantitative PCR.**
(TIF)

## Acknowledgments

We would like to give special thanks to the colleagues from the Department of Stomatology and its Nursing Department at Affiliated Hospital of Jinggangshan University provided technical guidance and data analysis services.

## Author Contributions

**Conceptualization:** Guanghua Xiong.

**Data curation:** Xi Li, Manni Luo.

**Formal analysis:** Xi Li, Manni Luo, Siwan Mao.

**Funding acquisition:** Guanghua Xiong.

**Investigation:** Siwan Mao, Ruiying Zhang, Juan Li.

**Methodology:** Xi Li, Ruiying Zhang, Ziwei Meng, Juan Li.

**Project administration:** Xinjun Liao.

**Software:** Ziwei Meng.

**Supervision:** Guanghua Xiong, Xinjun Liao.

**Writing – original draft:** Xinjun Liao.

**Writing – review & editing:** Guanghua Xiong.

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
