## [Decision Letter · Decision Letter 0]

17 Jun 2024

PONE-D-23-37049Peptidoglycan recognition protein PGRP-5 is involved in immune defense and neuro-behavioral disorders in zebrafish embryosPLOS ONE

Dear Dr. Liao,

Thank you for submitting your manuscript to PLOS ONE. After careful consideration, we feel that it has merit but does not fully meet PLOS ONE’s publication criteria as it currently stands. Therefore, we invite you to submit a revised version of the manuscript that addresses the points raised during the review process.

We look forward to receiving your revised manuscript.

Kind regards,

Dharmendra Kumar Meena

Academic Editor

PLOS ONE

Journal Requirements:

4. Thank you for stating the following in the Acknowledgments Section of your manuscript: "This work was financially supported by the National Natural Scientific Foundation of China (82160048), Anhui Natural Science Foundation Project (2308085MH265), Jiangxi Natural Science Foundation Project (20202ACBL215009) and Jiangxi Science and Technology Research Project of the Education Department (GJJ2201611).

Please remove any funding-related text from the manuscript and let us know how you would like to update your Funding Statement. Currently, your Funding Statement reads as follows: "National Natural Science Foundation of China".

Additional Editor Comments:

The article is recommended for major revision

Reviewers' comments:

Reviewer's Responses to Questions

**Comments to the Author**

1. Is the manuscript technically sound, and do the data support the conclusions?

Reviewer #1: Partly

Reviewer #2: Yes

2. Has the statistical analysis been performed appropriately and rigorously? 

Reviewer #1: No

Reviewer #2: Yes

3. Have the authors made all data underlying the findings in their manuscript fully available?

Reviewer #1: Yes

Reviewer #2: No

4. Is the manuscript presented in an intelligible fashion and written in standard English?

Reviewer #1: No

Reviewer #2: Yes

5. Review Comments to the Author

Reviewer #1: Xiong et al present an interesting finding that the Peptidoglycan recognition protein (PGRP) PGRP-5 is involved with zebrafish inflammatory response and locomotion. In particular the authors build a case for PGRP-5 through the use of sequence analysis, tissue expression levels with additional characterization of function with bacterial challenge (Gram + & -) and embryo tracking. However, the manuscript as presented needs several additional points of clarification in order to evaluate the findings thoroughly, as this reviewer’s background is pathogenesis in mammalian systems.

Major:

1) In line 90 the authors mentioned DrPGRP5 was cloned in the present study, but information on cloning is lacking in the methods. Moreover, in Fig4 the authors are measuring the mRNA level of DrPGRP5 – is this the endogenous PGRP5 or the short fragment that was cloned in response to bacterial challenge?

2) The authors should include methods for bacterial challenge and cultivation, as only the 107 CFU/mL is not enough. As it reads, it seems bacteria were added to embryos and over a 3-day period only embryos were checked for expression. Moreover, the authors mention dead embryos and larvae were removed daily – how many embryos died? How many embryos to larva events occur? This survival data may support the author's claims.

3) The authors use PGRP5-KD in fig5 to determine mRNA levels of inflammatory genes, but there is no mention of the stimulus, or if this is baseline. In the same context, what is fig6 (in the KD fish) measuring? Did the authors test these levels during bacterial challenge for both experimental setups? In line 248, please provide the data showing knockdown and define “pre-experimental” results.

4) Several points the authors go between embryo and larvae for experiments. Clarity is needed here – particularly for the 2.4 methods, fig3 and fig6.

5) Lines 327-329 needs context – exposed to what? PRRs recognize components of microbes.

Minor:

Several sections of the manuscript need revision for the reader’s clarity such as in lines 17-19, or 69-71.

Statistics across methods and the figure legends needs to be clarified as the methods mention n=4 and SEM were used, but in in several figures it is stated biological replicates and SD. Authors should check that the correct statistical test has been employed for the data (ie: are they parametric? )

Fig1 and results: Additional information pertaining to the % homology of the PGRPs is needed. Figure legend should include information about the full protein perhaps, as only the C-terminal amidase is shown.

Fig6: has n=4 from three biological replicates stated. Does this mean 4 samples per biological for a total n=12? Please show the individual data points and clarify this figure legend.

Reviewer #2: Manuscript Number: PONE-D-23-37049

Full Title: Peptidoglycan recognition protein PGRP-5 is involved in immune defense and neurobehavioral disorders in zebrafish embryos

Reviewer’s comments

A. Title (Line 1-2)

1. Appropriately written. Replace s with c in defence

B. Abstract –

1. Concise and reflect the content of the works done.

2. Line (27-28) – Cytokines are italicised here. It should be followed in the entire manuscript.

C. Keywords

1. Selection of keywords reflect the content completely.

D. Introduction

1. Line 50-51- Write immune response against instead of of .

2. Line 51 – Write full form of IMD

3. Line 63 - and structure also indicates. Add its before structure

4. Line 66 - challenge of pathogenic. Write due to instead of of. Add further before The

5. Line-75 - I should be small in intracellular

6. Line-81 – development and behaviour, of whom?

7. Line-83 – microbiota-brain interactions, of whom?

8. Line-85 – development and behaviour, of whom?

9. Line-85-87 – In…sentence is not complete.

10. Line 93- remove further after we

11. Line-96- replace aquatic ecosystem with fish

E. Materials and Methods

1. Line-142 – Write complete name of E. coli and M. lutues and italicize it.

2. Line-143 - 1 x 107 CFU/mL. it should be superscripted.

3. Line-162 - -actin…??

4. Line-168&181 – 500 L…??

F. Results

1. Line – 233 – write potential instead of potentially

2. Line-241- write Gram - before positive

3. Line-257 – TNF- α

4. Line-281- locomotor not loco-motor

G. Discussions

1. Written well.

H. Conclusion

1. Written well

6. PLOS authors have the option to publish the peer review history of their article (what does this mean?). If published, this will include your full peer review and any attached files.

Reviewer #1: No

Reviewer #2: No

---

## [Author Response · Author response to Decision Letter 0]

29 Sep 2024

Journal Requirements:

1.When submitting your revision, we need you to address these additional requirements. Please ensure that your manuscript meets PLOS ONE's style requirements, including those for file naming. 

Answer: We have made modifications to the manuscript format according to the requirements of PLOS ONE, including citation of references and file naming.

2.Did you know that depositing data in a repository is associated with up to a 25% citation advantage ? If you’ve not already done so, consider depositing your raw data in a repository to ensure your work is read, appreciated and cited by the largest possible audience. 

Answer: Thank you for your reminder. This manuscript does not have any original data that needs to be stored in a public database. We have included the original amino acid sequence for constructing the phylogenetic tree in the supplementary document.

3.We note that the grant information you provided in the ‘Funding Information’ and ‘Financial Disclosure’ sections do not match. 

Answer: Thank you for your reminder. We have revised the Funding Information section in the manuscript.

4.We note that you have provided funding information that is not currently declared in your Funding Statement. However, funding information should not appear in the Acknowledgments section or other areas of your manuscript. We will only publish funding information present in the Funding Statement section of the online submission form. Please remove any funding-related text from the manuscript and let us know how you would like to update your Funding Statement. Currently, your Funding Statement reads as follows: "National Natural Science Foundation of China". Please include your amended statements within your cover letter; we will change the online submission form on your behalf.

Answer: Thank you for your suggestions. We have deleted the funding information in the manuscript. Our amended funding statements as follows: “This research was funded by the National Natural Scientific Foundation of China (82160048), Anhui Natural Science Foundation Project (2308085MH265), Anhui Science and Technology Research Project of the Education Department (2024AH040205), Anhui Excellent Talents Support Program for Universities (YQYB20230170), Jiangxi Science and Technology Research Project of the Education Department (GJJ2201611) and PhD Initiation Project of Fuyang Normal University (KYQD20230004).” Please change the online submission form on my behalf.

5. PLOS ONE now requires that authors provide the original uncropped and unadjusted images underlying all blot or gel results reported in a submission’s figures or Supporting Information files. In your cover letter, please note whether your blot/gel image data are in Supporting Information or posted at a public data repository, provide the repository URL if relevant, and provide specific details as to which raw blot/gel images, if any, are not available. Email us at plosone@plos.org if you have any questions.

Answer: Thank you for your reminding. In our revised version, the results and supplementary documents did not contain any blot or gel images.

6.We note that you have included the phrase “data not shown” in your manuscript. Unfortunately, this does not meet our data sharing requirements. PLOS does not permit references to inaccessible data. We require that authors provide all relevant data within the paper, Supporting Information files, or in an acceptable, public repository. 

Answer: Thank you for your suggestion. We have deleted the phrase “data not shown” in our revised manuscript. Meanwhile, we have added a supplementary document in the corresponding area to support our conclusion.

7.Please include captions for your Supporting Information files at the end of your manuscript, and update any in-text citations to match accordingly. Please see our Supporting Information guidelines for more information: http://journals.plos.org/plosone/s/supporting-information.

Answer: Thank you for your suggestion. We have added the captions for my Supporting Information files at the end of the revised manuscript.

Reviewer #1: Xiong et al present an interesting finding that the Peptidoglycan recognition protein (PGRP) PGRP-5 is involved with zebrafish inflammatory response and locomotion. In particular the authors build a case for PGRP-5 through the use of sequence analysis, tissue expression levels with additional characterization of function with bacterial challenge (Gram + & -) and embryo tracking. However, the manuscript as presented needs several additional points of clarification in order to evaluate the findings thoroughly, as this reviewer’s background is pathogenesis in mammalian systems.

Answer: Thank you for your positive comments on our manuscript. In order to improve the whole quality of the manuscript, we have modified and polished many sections of the full text, and we have also supplemented several experiment data and corrected some mistakes in grammar on our previous draft.

Major:

1)In line 90 the authors mentioned DrPGRP5 was cloned in the present study, but information on cloning is lacking in the methods. Moreover, in Fig4 the authors are measuring the mRNA level of DrPGRP5 – is this the endogenous PGRP5 or the short fragment that was cloned in response to bacterial challenge?

Answer: Thank you for your suggestions, it was indeed our mistake and lack of clarity in expression. We did not clone the DrPGRP5 gene in vitro cells, but only using specific DrPGRP5 gene primers to detect partial fragments of this gene by conducting quantitative PCR experiments. We have made modifications and improvements to this section of the manuscript. Moreover, in Fig4, the control group was the dynamic expression level of endogenous DrPGRP5 gene in zebrafish embryos at four key time points, while the treatment group was the mRNA expression level of DrPGRP5 gene in zebrafish embryos stimulated by E. coli and M. luteus bacteria, respectively. This experiment was also conducted through real-time quantitative PCR detection using specific DrPGRP5 gene primers.

2)The authors should include methods for bacterial challenge and cultivation, as only the 107 CFU/mL is not enough. As it reads, it seems bacteria were added to embryos and over a 3-day period only embryos were checked for expression. Moreover, the authors mention dead embryos and larvae were removed daily – how many embryos died? How many embryos to larva events occur? This survival data may support the author's claims.

Answer: Thank you for your suggestions. We have added detailed methods for bacterial culture and zebrafish embryo stimulation in the methods section, and we have also included survival rate and hatching rate data in Fig. 4A and Fig. 4B.

3)The authors use PGRP5-KD in fig5 to determine mRNA levels of inflammatory genes, but there is no mention of the stimulus, or if this is baseline. In the same context, what is fig6 (in the KD fish) measuring? Did the authors test these levels during bacterial challenge for both experimental setups? In line 248, please provide the data showing knockdown and define “pre-experimental” results.

Answer: Thank you for your attention. Based on previous research data, we have found that the DrPGRP5 gene exhibits unique characteristics in terms of evolution and tissue differential expression. Therefore, we want to explore how the expression of relevant inflammatory cytokines changes under the condition of DrPGRP5 gene knock-down. So we did not detect the expression changes of these inflammatory genes under two bacterial stimulation conditions in Fig. 5. Similarly, in Fig. 6, we only examined the behavior and enzyme activity changes compared to normal control zebrafish embryos under DrPGRP5 gene knock-down conditions. In addition, we provided the knockdown efficiency of the PGRP5 gene in the supplementary files.

4)Several points the authors go between embryo and larvae for experiments. Clarity is needed here – particularly for the 2.4 methods, fig3 and fig6.

Answer: Thanks for your nice suggestions. We have clarified the detection time (zebrafish embryos at 72 hpf) in the 2.4 methods, fig3 and fig6, respectively.

5)Lines 327-329 needs context – exposed to what? PRRs recognize components of microbes.

Answer: Thanks for your nice suggestions. IAB and probiotic treatments promote neuronal health and influence inflammatory pathways through PGRPs-mediated neural and immune signaling in drosophila model. We revised this information in the discussion section.

Minor:

Several sections of the manuscript need revision for the reader’s clarity such as in lines 17-19, or 69-71.

Answer: Thanks for your nice suggestions. We have made modifications to these parts in an effort to make our expression clearer.

Statistics across methods and the figure legends needs to be clarified as the methods mention n=4 and SEM were used, but in in several figures it is stated biological replicates and SD. Authors should check that the correct statistical test has been employed for the data (ie: are they parametric? )

Answer: Thanks for your nice suggestions. We check the statistical methods of all data to ensure consistency.

Fig1 and results: Additional information pertaining to the % homology of the PGRPs is needed. Figure legend should include information about the full protein perhaps, as only the C-terminal amidase is shown.

Answer: Thanks for your nice suggestions. We have added the homology percentage data of PGRPs sequences in Fig. 1B.

Fig6: has n=4 from three biological replicates stated. Does this mean 4 samples per biological for a total n=12? Please show the individual data points and clarify this figure legend.

Answer: Thanks for your nice suggestions. We have added the homology percentage data of PGRPs sequences in Fig. 1B. Regarding enzyme activity data, each sample contains 4 biological replicates, and each biological replicate contains 3 technical replicates. All data were subjected to statistical analysis using SPSS software.

Reviewer #2: Reviewer’s comments

A. Title (Line 1-2)

1. Appropriately written. Replace s with c in defence

Answer: Thank you for good suggestion. We have revised the word “defense” to “defence” in the manuscript.

B. Abstract –

1. Concise and reflect the content of the works done.

Answer: Thank you for your suggestion. We have polished and reduced the content of the abstract section.

2.Line (27-28) – Cytokines are italicised here. It should be followed in the entire manuscript.

Answer: Thank you. Cytokines are italicised in the entire revised manuscript.

C. Keywords

1. Selection of keywords reflect the content completely.

Answer: Thanks. We have re-selected and modified the keywords to better reflect the main research content of the entire article. 

D. Introduction

1. Line 50-51- Write immune response against instead of of.

Answer: We have made corresponding revisions according to your nice suggestions.

2.Line 51 – Write full form of IMD

Answer: The full name of IMD is immune deficiency (IMD).

3.Line 63 - and structure also indicates. Add its before structure

Answer: We have revised it according to your suggestions.

4.Line 66 - challenge of pathogenic. Write due to instead of of. Add further before The

Answer: We have revised it according to your suggestions.

5.Line-75 - I should be small in intracellular

Answer: We have revised it according to your suggestions.

6.Line-81 – development and behaviour, of whom?

Answer: In neurons and glial cells.

7.Line-83 – microbiota-brain interactions, of whom?

Answer: In the placenta and developing brain.

8.Line-85 – development and behaviour, of whom?

Answer: Autism Spectrum Disorder (ASD).

9.Line-85-87 – In…sentence is not complete.

Answer: We have already deleted this sentence in the revised manuscript.

10. Line 93- remove further after we

Answer: We have revised it according to your suggestions.

11.Line-96- replace aquatic ecosystem with fish

Answer: We have revised it according to your suggestions.

E. Materials and Methods

1. Line-142 – Write complete name of E. coli and M. lutues and italicize it.

Answer: Escherichia coli (E. coli) and Micrococcus luteus (M. luteus).

3.Line-143 - 1 x 107 CFU/mL. it should be superscripted.

Answer: We have already superscripted it in the revised manuscript.

4.Line-162 - -actin…??

Answer: It is β-actin, we revised it in the manuscript.

5.Line-168&181 – 500 L…??

Answer: It is 500 µl, we revised it in the manuscript.

F. Results

1. Line – 233 – write potential instead of potentially

Answer: We have made corresponding revision in the manuscript.

2.Line-241- write Gram - before positive

Answer: We have made corresponding revision in the manuscript.

3.Line-257 – TNF- α

Answer: We have made corresponding revision in the manuscript.

4.Line-281- locomotor not loco-motor

Answer: We have made corresponding revision in the manuscript.

G. Discussions

1. Written well.

Answer: Thank you for your positive feedback. We will further revise and improve the discussion section.

H. Conclusion

1. Written well

Answer: Thank you for your positive feedback. We will further revise and improve the conclusion section.

---

## [Decision Letter · Decision Letter 1]

26 Nov 2024

PONE-D-23-37049R1Peptidoglycan recognition protein PGRP-5 is involved in immune defence and neuro-behavioral disorders in zebrafish embryosPLOS ONE

Dear Dr. Liao,

Thank you for submitting your manuscript to PLOS ONE. After careful consideration, we feel that it has merit but does not fully meet PLOS ONE’s publication criteria as it currently stands. Therefore, we invite you to submit a revised version of the manuscript that addresses the points raised during the review process.

We look forward to receiving your revised manuscript.

Kind regards,

Jisheng Liu, Ph.D.

Academic Editor

PLOS ONE

Journal Requirements:

Additional Editor Comments:

The manuscript has been revised accordingly. The authors have addressed all the comments from the reviewers. However, there are some details that the authors should pay attention to. Therefore, I invite the authors to revise the details before the manuscript is accepted.

1. Beside the biological replicates, the technical replicates should be mentioned in M&M and the captions, such as Fig. 3, Fig. 4, Fig. 5, Fig. 6.

2. In Fig. 3A, what does the green and pink boxes stand for? Please indicates in the text and captions. Besides, the PGRP domain should be boxed as well.

3. Please explain and discuss why AChE was up-regulated in PGRP5-KD?

Reviewers' comments:

Reviewer's Responses to Questions

**Comments to the Author**

1. If the authors have adequately addressed your comments raised in a previous round of review and you feel that this manuscript is now acceptable for publication, you may indicate that here to bypass the “Comments to the Author” section, enter your conflict of interest statement in the “Confidential to Editor” section, and submit your "Accept" recommendation.

Reviewer #1: All comments have been addressed

Reviewer #2: All comments have been addressed

2. Is the manuscript technically sound, and do the data support the conclusions?

Reviewer #1: Yes

Reviewer #2: Yes

3. Has the statistical analysis been performed appropriately and rigorously? 

Reviewer #1: Yes

Reviewer #2: Yes

4. Have the authors made all data underlying the findings in their manuscript fully available?

Reviewer #1: Yes

Reviewer #2: Yes

5. Is the manuscript presented in an intelligible fashion and written in standard English?

Reviewer #1: Yes

Reviewer #2: Yes

6. Review Comments to the Author

Reviewer #1: I appreciate the diligence of the authors to address my points in the previous submission, and now feel these works are ready for publication. I do have a few minor comments below:

Minor:

- Fig3 B the intestine x-axis line is not straight

- Line 146, suggest to revise “in a nutshell…” vernacular to something like “Briefly,”

- Fig4 B y-axis label revise – states “Percent \\ %”

- It would be interesting to challenge PGRP-5 KD fish vs scrambled with E coli or M luteus as a definitive demonstration that loss of PGRP-5 leads to even more killing and reduced rate of hatching in fig4

- In 3.3 and throughout the manuscript be consistent with stating “Gram positive” / “Gram negative” or “Gram-positive” / “Gram-negative”

Reviewer #2: Dear Authors,

You have made necessary changes in revised manuscript satisfactorily which is technically sound, and data support the conclusions, data has been anaysed correctly and well written in standard English.

7. PLOS authors have the option to publish the peer review history of their article (what does this mean?). If published, this will include your full peer review and any attached files.

Reviewer #1: No

Reviewer #2: **Yes: **RAJU BAITHA

---

## [Author Response · Author response to Decision Letter 1]

28 Nov 2024

Additional Editor Comments:

The manuscript has been revised accordingly. The authors have addressed all the comments from the reviewers. However, there are some details that the authors should pay attention to. Therefore, I invite the authors to revise the details before the manuscript is accepted.

1. Beside the biological replicates, the technical replicates should be mentioned in M&M and the captions, such as Fig. 3, Fig. 4, Fig. 5, Fig. 6.

Answer: Thank you for your suggestions. Each value contains at least four biological replicates, and each biological replicate contains three technical replicates. We have added these information in the Fig. 3, Fig. 4, Fig. 5, Fig. 6 and M&M sections.

2. In Fig. 3A, what does the green and pink boxes stand for? Please indicates in the text and captions. Besides, the PGRP domain should be boxed as well.

Answer: Thank you for your suggestions. The green boxes represent the low complexity region, and pink boxes stand for PGRP domains that contained a conserved N-acetylmuramoyl-L-alanine amidase (ami_2) region. We have added these information in the legend of Fig. 3A.

3. Please explain and discuss why AChE was up-regulated in PGRP5-KD?

Answer: Thank you for your suggestions. The up-regulation of Acetylcholinesterase (AChE) in PGRP5 knockdown (KD) could potentially be attributed to several factors. One possibility is that the knockdown of PGRP5 might disrupt normal cellular signaling pathways that regulate the expression of AChE. PGRP5 could be involved in a complex network of interactions and its knockdown could lead to compensatory mechanisms or imbalances in downstream signaling, resulting in the increased expression of AChE. Another consideration is that PGRP5 might directly or indirectly interact with transcription factors or regulatory elements that control the expression of the AChE gene. The absence or reduced function of PGRP5 could remove an inhibitory effect or trigger a feedback loop that promotes AChE up-regulation.

Reviewer #1: I appreciate the diligence of the authors to address my points in the previous submission, and now feel these works are ready for publication. I do have a few minor comments below:

Minor:

- Fig3 B the intestine x-axis line is not straight

Answer: Thank you for your reminding. We have revised the intestine x-axis in Fig. 3B.

- Line 146, suggest to revise “in a nutshell…” vernacular to something like “Briefly,”

Answer: Thank you for your reminding. We have made the corresponding revisions according to your suggestions.

- Fig4 B y-axis label revise – states “Percent \\ %”

Answer: Thank you for your reminding. We have made the corresponding revisions according to your suggestions.

- It would be interesting to challenge PGRP-5 KD fish vs scrambled with E coli or M luteus as a definitive demonstration that loss of PGRP-5 leads to even more killing and reduced rate of hatching in fig4

Answer: Thank you for your suggestions. What we want to demonstrate in Fig. 4 is that the Dr-PGRP5 gene exhibits different expression patterns under the stimulation of Gram-positive and Gram-negative bacteria, respectively. Meanwhile, M.luteus bacteria had lower survival and hatching rates compared to E. coli bacteria on normally developing zebrafish embryos. Actually, we did not test whether PGRP5-KD fish resulted in lower survival and hatching rates compared to normal fish, which is indeed an interesting question and we will explore this issue in future studies.

- In 3.3 and throughout the manuscript be consistent with stating “Gram positive” / “Gram negative” or “Gram-positive” / “Gram-negative”

Answer: Thank you for your suggestions. “Gram-positive” / “Gram-negative” were consistent throughout the entire manuscript.

---

## [Editor Report · Decision Letter 2]

1 Dec 2024

Peptidoglycan recognition protein PGRP-5 is involved in immune defence and neuro-behavioral disorders in zebrafish embryos

PONE-D-23-37049R2

Dear Dr. Liao,

We’re pleased to inform you that your manuscript has been judged scientifically suitable for publication and will be formally accepted for publication once it meets all outstanding technical requirements.

Kind regards,

Jisheng Liu, Ph.D.

Academic Editor

PLOS ONE

---

## [Editor Report · Acceptance letter]

16 Dec 2024

PONE-D-23-37049R2 

PLOS ONE

Dear Dr. Liao, 

I'm pleased to inform you that your manuscript has been deemed suitable for publication in PLOS ONE. Congratulations! Your manuscript is now being handed over to our production team.

Kind regards, 

on behalf of

Professor Jisheng Liu 

Academic Editor

PLOS ONE